# CAST: Concurrent Recognition and Segmentation with Adaptive Segment Tokens

## Abstract

Recognizing an image and segmenting it into coherent regions are often treated as separate tasks. Human vision, however, has a general sense of segmentation hierarchy before recognition occurs. We are thus inspired to learn image recognition with hierarchical image segmentation based entirely on unlabeled images. Our insight is to learn fine-to-coarse features concurrently at superpixels, segments, and full image levels, enforcing consistency induced segmentations while maximizing discrimination among image instances.

Our model innovates vision transformers in three aspects. **1)** We use adaptive segment tokens instead of fixed-shape patch tokens. **2)** We create a token hierarchy by inserting graph pooling between transformer blocks, naturally producing consistent multi-scale segmentations while increasing the segment size and reducing the number of tokens. **3)** We produce hierarchical image segmentation for free *while* training for recognition by maximizing image-wise discrimination.

Our work delivers the first concurrent recognition and hierarchical segmentation model *without any supervision*. Validated on ImageNet and PASCAL VOC, it achieves better recognition and segmentation with higher computational efficiency.

## 1 Introduction

Convolutional neural networks (CNN) (LeCun et al., 1989; Krizhevsky et al., 2012; He et al., 2016) and Vision Transformers (ViT) (Dosovitskiy et al., 2020) have been very successful in computer vision. However, recognizing an image and segmenting it into coherent regions are treated as separate tasks or learned sequentially (Martin et al., 2001). Fig. 1 illustrates a common practice: CNN (ViT) predicts the semantic class of an image based on the image-level feature from the output of the final convolutional layer (transformer block), and additional clustering based on earlier pixel-wise features is required to generate image segmentation (Hwang et al., 2019; Ke et al., 2022).

However, human vision has a general sense of segmentation hierarchy, in terms of groups of pixels or *segments*, before recognition even occurs. This perceptual organization perspective (Witkin & Tenenbaum, 1983; Biederman, 1987) has been overlooked in CNN and ViT architectures: models optimized for image classification tend to latch onto discriminative parts (Selvaraju et al., 2017) such as faces, often missing inconspicuous body parts that go with the face. Previous methods seldom model how different parts such as *face* and *body* are organized for the whole *animal* explicitly.

To understand the connections between parts and wholes, visual information must be extracted locally and globally. There are three major approaches (Fig. 2):

1. **Spatial downsampling:** With pixels laid on a regular grid, features are extracted from patches. The granularity of visual information is determined by the patch size. Max or mean pooling (Krizhevsky et al., 2012; Simonyan & Zisserman, 2014), uniform sub-sampling (striding) (He et al., 2016) and patch merging (Liu et al., 2021) are performed multiple times to increase the effective receptive field size (Luo et al., 2016). To generate a segmentation, the output features need to be upsampled and clustered (e.g. K-Means (Hwang et al., 2019)), often resulting in over-smoothed boundaries.

2. **Attention:** Inspired by Natural Language Processing (NLP), image patches are treated as visual word *tokens* of the entire image *document*. To extract more global information, ViT contextually updates feature representations based on pair-wise correlation among all the tokens of an image,

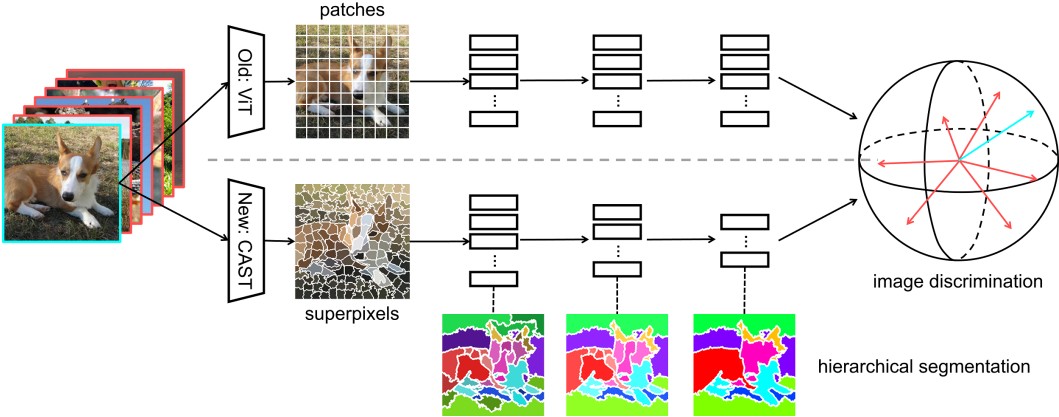

Figure 1: We innovate vision transformer models to concurrently learn image recognition and hierarchical image segmentation from unlabeled images alone. **Top:** ViT (Dosovitskiy et al., 2020) takes patch tokens as inputs and maintains the same large number of tokens through all encoder blocks. Image segmentation would require additional pixel-wise clustering (e.g. K-Means) on the fixed patch-wise features. **Bottom**: Our model takes segment tokens as inputs and hierarchically groups them into fewer coarsened region tokens. Unlike patch tokens, these segment tokes adapt to the image and vary in shape. We unify fine-to-coarse feature learning at multiple levels in a single model to support not only recognition with maximum image-wise discrimination, but also segmentation with consistency across the hierarchy. Consequently, we achieve better recognition and segmentation with higher computational efficiency.

  using attention modules (Vaswani et al., 2017). However, ViT is computationally inefficient as all image tokens are kept in every transformer block.

3. **Significance-based subsampling:** To increase ViT's computational efficiency, tokens are subsampled at higher levels based on their significance scores. PoWER-BERT (Goyal et al., 2020) and Token Pooling (Marin et al., 2021) define the *significance score* as the total attention given to each token from all other tokens. Downsampling then retains only the most dominant visual features in the image. Such methods only keep the most informative tokens in final output representations.

These existing methods have two major issues. **1)** Both CNN and ViT models take regularly shaped patch features as inputs, regardless of what is in the image. Image segmentation derived from such representations often fails to align with contours. **2)** Image segmentation does not involve local-to-global feature extraction, which is treated as a separate visual task from image-wise recognition.

Our first insight is that pixel groupings are not a computational inconvenience (as opposed to regular patches), but a natural structure to be exploited for better visual computing. Unlike existing CNN and ViT which extract features on a regular grid throughout the entire model, we directly get to low-level pixel groupings at an early stage and develop feature representations subsequently. Our model takes segment features as input tokens and carries this adaptive segment representation through deeper layers. Post-processing with pixel-wise clustering methods is no longer needed.

Our second insight is to derive fine-to-coarse pixel groupings jointly with local-to-global feature extraction. Given a set of token features, we cluster them into fewer components. The next-level feature is the result of pooling current features within each cluster. Since our input tokens come from segments of an image, feature clustering turns fine-grained segments into coarse-grained regions. By repeating the procedure, we obtain a consistent fine-to-coarse (hierarchical) image segmentation and corresponding feature representations at each level of granularity.

We propose to integrate such data-driven perceptual organization into Vision Transformers (Dosovitskiy et al., 2020). We develop *Concurrent recognition and segmentation with Adaptive Segment Tokens* (CAST). It has three novel aspects (Fig. 2). **1)** We use adaptive segment tokens instead of fixed-shape patch tokens. They no longer live on a regular grid, and their shapes and numbers vary with the image. **2)** We create a token hierarchy by inserting graph pooling between transformer blocks, naturally producing consistent multi-scale segmentations while increasing the segment size

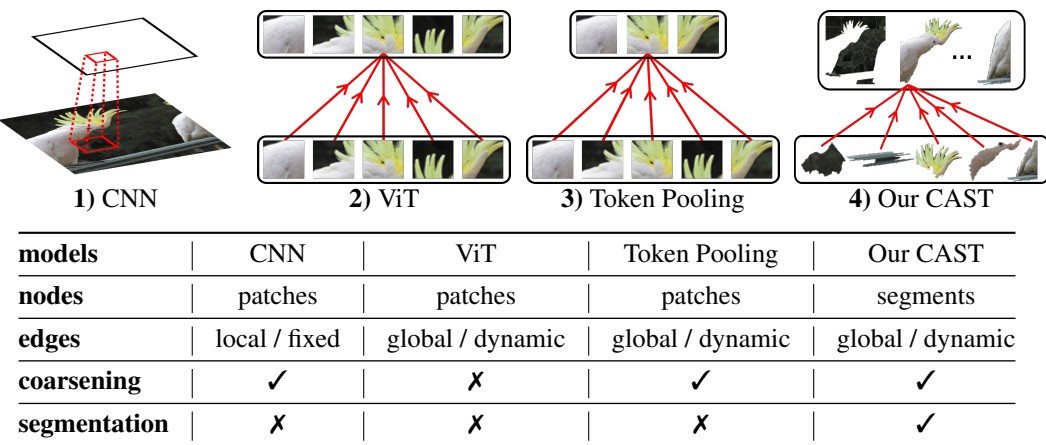

| models | CNN | ViT | Token Pooling | Our CAST |
|---|---|---|---|---|
| **nodes** | patches | patches | patches | segments |
| **edges** | local / fixed | global / dynamic | global / dynamic | global / dynamic |
| **coarsening** | ✓ | ✗ | ✓ | ✓ |
| **segmentation** | ✗ | ✗ | ✗ | ✓ |

Figure 2: Our model bootstraps from low-level segment tokens, coarsens visual information by clustering, and merges fine-grained segment tokens into coarser-grained region tokens. **Top)** We compare different models in what they operate on and how they extract local-to-global information. CNN (Krizhevsky et al., 2012; He et al., 2016) computes features on a regular grid and handles more global information by spatial downsampling. ViT (Dosovitskiy et al., 2020) takes regularly shaped patches as inputs, and updates features using attention (Vaswani et al., 2017). Token Pooling (Marin et al., 2021) subsamples tokens by their significance scores. Our CAST takes segment tokens as inputs and coarsens them into larger region tokens, which adapt to the image. **Bottom)** We compare models from a graph perspective, in terms of nodes, edges (connections between nodes), and whether the graph coarsening is used and segmentation is produced. Our method is the only one that uses adaptive segment tokens with coarsening and outputs segmentations.

and reducing the number of tokens. **3)** We learn segmentation for free *while* training the model for unsupervised recognition by maximizing image-wise discrimination (Chen et al., 2021). Neither recognition nor segmentation requires any labeled supervision.

Our experimental results demonstrate our superior computational efficiency and segmentation accuracy on ImageNet and PASCAL VOC. More importantly, our model delivers far more precise foreground masks which can be very useful in a wide range of dense pixel applications.

In short, our work makes three major contributions. **1)** We develop the first vision transformer model that can *concurrently* achieve image-wise recognition and hierarchical image segmentation without any additional processing. **2)** We outperform existing token coarsening methods for both image classification and segmentation tasks. We achieve a better trade-off between model efficiency and task performance. **3)** We deliver better attention maps that capture foreground semantics without supervision, with many potential applications beyond segmentation.

## 2    RELATED WORKS

**Vision Transformers.** Vision transformers (ViT) (Dosovitskiy et al., 2020) and its followups (Touvron et al., 2021; Yuan et al., 2021; Wang et al., 2021a) adopt the transformer architecture originally for NLP proposed in Vaswani et al. (2017). ViT achieves remarking performance on image classification (Deng et al., 2009), however, its high computation costs limit its applications. Its computational complexity is derived from two factors: the latent feature dimensions and the number of tokens.

To reduce the latent feature computation, one direction is to restrict the attention connections by leveraging spatial relationships in the data (Parmar et al., 2018; Ramachandran et al., 2019; Beltagy et al., 2020; Child et al., 2019; Zaheer et al., 2020) or utilizing hashing, sorting, or compression (Kitaev et al., 2020; Vyas et al., 2020; Tay et al., 2020; Liu et al., 2018; Wang et al., 2020). To reduce the number of tokens, two camps of approaches are proposed. The first is to apply the concepts of hierarchical convolutional neural nets to downsample tokens using various pooling methods (Liu et al., 2021; Heo et al., 2021; Dong et al., 2022). The other attempts to measure the significance scores among the tokens and drop or prune tokens accordingly (Goyal et al., 2020; Rao et al., 2021;

Marin et al., 2021). This camp is the most related to our work. Our work differs in that tokens are not discarded but merged into coarser ones.

TCFormer (Zeng et al., 2022) and GroupViT (Xu et al., 2022) are two recently proposed hierarchical vision transformers. Both models still use patch tokens. TCFormer is only applied to supervised tasks, and GroupViT requires text supervision. Our work operates on adaptive segment tokens, which naturally induce hierarchical image segmentations. Our model does not require any human label.

**Superpixels.** Superpixels are referred to as sets of locally connected pixels that contain coherent structures (e.g. colors) (Ren & Malik, 2003). Superpixels have been applied to densely labeling tasks, such as body-part parsing (Mori et al., 2004), saliency detection (Ren et al., 2013), image segmentation (Gould et al., 2008; Fulkerson et al., 2009; Sharma et al., 2014; Gadde et al., 2016), and hierarchical segmentation (Wei et al., 2018). Recently, Zhang et al. (2022) tackle semantic segmentation by replacing patch with superpixel tokens in ViT architectures. In contrast, our model further creates a segment hierarchy and performs both classification and segmentation concurrently.

**Image Segmentation and Clustering.** Image segmentation is referred to as partitioning an image into coherent regions. Classic methods have two steps: extracting local features and clustering them based on different criteria, e.g., mode-finding (Comaniciu & Meer, 2002; Banerjee et al., 2005), or graph partitioning (Felzenszwalb & Huttenlocher, 2004; Shi & Malik, 2000; Malik et al., 2001; Yu & Shi, 2003a; 2004). A hierarchical segmentation is predicted as output for comparing against human perception (Arbelaez et al., 2010). The common approaches, to avoid ambiguities along object boundaries, typically resort to contour detection (Hwang & Liu, 2015; Xie & Tu, 2015) and eliminate contours iteratively to form multi-scale segmentations (Arbelaez et al., 2010). Such approaches train on the finest level of ground-truth segmentation and hope to produce coarser levels of segmentation automatically for inference. Our work operates directly on segments as opposed to contour proxies.

**Concurrent Recognition and Segmentation.** This idea was explored before the deep learning era: Recognition by grouping compatible patches and segmentation by grouping visually similar pixels are solved together through detected pixel-patch relations, resulting in object-specific segmentation (Yu et al., 2002; Yu & Shi, 2003b) and figure-ground segmentation (Maire, 2010; Maire et al., 2011). These methods rely on not only handcrafted features and grouping cues, but also pre-trained object part detectors, whereas our work does not use any such priors or supervised training.

**Self-supervised Segmentation and Representation Learning.** Recent works can be categorized into three camps. **1)** A straightforward approach is to leverage self-supervised image recognition and transfer the model to segmentation by increasing the location sensitivity (Wu et al., 2018; He et al., 2020; Chen et al., 2020; Wang et al., 2021c), adding a contrastive loss across views (Wang et al., 2021b), or by stronger augmentation and constrained cropping (Van Gansbeke et al., 2021; Selvaraju et al., 2021). **2)** A pixel-wise cluster predictor can be learned by maximizing the mutual information between cluster predictions on augmented views of the same instance at corresponding pixels (Ji et al., 2019; Ouali et al., 2020). **3)** A pixel-level feature encoder can be learned directly by maximizing discrimination between pixels based on either contour-induced segments (Hwang et al., 2019), pre-computed region hierarchies (Zhang & Maire, 2020), or hierarchical clustering transformers (Ke et al., 2022). Segmentation is thus derived from pixel feature similarities. Our work unifies the first and the third camp, as we train ViT with a self-supervised image recognition framework while naturally producing unsupervised hierarchical segmentation.

## 3 CONCURRENT RECOGNITION AND HIERARCHICAL SEGMENTATION

At the core of our method is to consider image recognition and segmentation as concurrent, not separate, tasks. The basic idea is to extract local-to-global visual information by producing fine-to-coarse image segmentations and corresponding feature representations. Meanwhile, such multi-scale feature representations should support final image-level recognition.

We ground our approach on a general perspective: images as graphs where pixels are nodes. We first generate low-level superpixels given an input image and extract corresponding segment tokens. Then we integrate image segmentation into the ViT architecture, where transformer blocks and our proposed graph pooling modules take arbitrarily shaped segment, not fixed-shape patch, tokens as inputs and outputs coarsened segment tokens for the next transformer block. In other words, we augment ViT to cluster fine-grained segments into coarse-grained regions w.r.t group-wise correlation

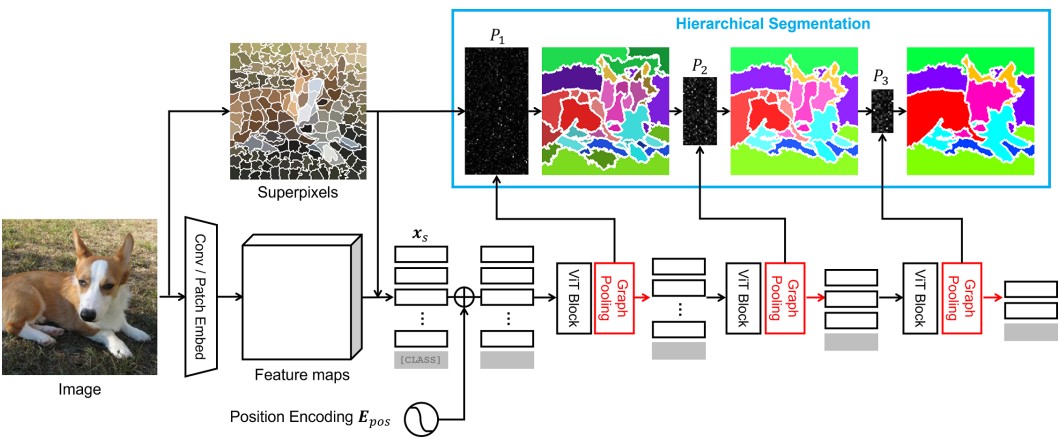

Figure 3: Our CAST jointly produces **1) a hierarchical image segmentation** and **2) corresponding multi-level features** for an input image. Building on ViT, our model operates on segment, not patch, tokens. We oversegment the image into superpixels, and extract initial segment tokens $\mathbf{x}_s$. We concatenate $\mathbf{x}_s$ with a `[CLASS]` token, and sum together with position encoding $\mathbf{E}_{pos}$ as inputs to subsequent transformer blocks. Followed by a **graph pooling** module, we group segments into coarser regions and aggregate features within each group. Let $P_l$ indicate how fine segments map to coarse regions at level $l$. The coarsened tokens become the inputs of next-level encoder blocks. Repeating the procedure, we generate a hierarchical image segmentation and multi-level features.

in the token feature space, resulting in hierarchical segmentation. By doing so, our model performs classification and hierarchical segmentation concurrently and more efficiently.

We describe the three components in our model in each subsection. **1)** Oversegmentations of input images based on low-level visual cues. **2)** Transformer encoder blocks to update token features. **3)** Graph pooling modules to cluster segment tokens and generate next-level representations. See Fig. 3.

## 3.1 THE FINEST-LEVEL PIXEL GROUPINGS AND TOKEN FEATURES

Existing methods tackle image segmentation separately from feature extraction. Such models extract patch features from the input image and then generate after-the-fact image segmentation by clustering the fixed patch features (similar to SegSort (Hwang et al., 2019) and HSG (Ke et al., 2022)). In stark contrast, our core idea is to involve image segmentation in feature extraction in the model architecture. We derive pixel groupings at an early stage, where we extract corresponding features for every segment. Our model carries such segment features to the subsequent layers and obtains image segmentations directly from the segment index of each pixel. Post-processing is thus not needed.

Given an input image, we start with the finest-level pixel groupings. We perform pixel groupings based on low-level visual cues to align segmentations with image contours. Specifically, we apply oversegmentation methods, e.g. Seeds (Bergh et al., 2012), to partition an image into locally connected and color-wise coherent regions–superpixels. Detailed in section 3.3, we can produce a hierarchical image segmentation with precisely localized contours by progressively grouping these superpixels.

To extract features of the superpixels, we first convolve the image with multiple convolutional layers, resulting in overlapping patch features. We then average pool those patch features within each superpixel to derive initial segment tokens $\mathbf{x}_s$ (with dimensions of the number of superpixels and number of feature channels).

We then input the initial segment tokens $\mathbf{x}_s$, along with additional priors to our model. Following ViT (Dosovitskiy et al., 2020), we append $\mathbf{x}_s$ with a learnable embedding (`[CLASS] token` $\mathbf{x}_{class}$) to encode the most prominent features of an image. $\mathbf{x}_{class}$ is randomly initialized and shared among different input images. We also enforce a spatial prior by adding $\mathbf{x}_s$ with relative position encodings $\mathbf{E}_{pos}$. We initiate $\mathbf{E}_{pos}$ at the same resolution as the convolutional patch features and then average pool within each superpixel. To sum up, the input segment token is $\mathbf{z}_0 = [\mathbf{x}_{class}; \mathbf{x}_s] + \mathbf{E}_{pos}$.

**Algorithm 1:** GraphPool

**Input:** Feature $\hat{\mathbf{z}}$ and number of clusters $n$.
**Output:** Coarsened feature $\mathbf{z}$ and assignments $P$
/* Sample n centroids. */
Centroid indices $S \leftarrow \mathrm{FPS}(\hat{\mathbf{z}})$
/* Refine features to encode
   correlation. */
Refined feature $\mathbf{u} \leftarrow \mathrm{MSA}(\mathrm{LN}(\hat{\mathbf{z}})) + \hat{\mathbf{z}}$
Normalized feature $\mathbf{u} \leftarrow \mathbf{u} - \mathrm{mean}(\mathbf{u}) + \mathrm{bias}$
Centroid feature $\mathbf{v} \leftarrow \{\mathbf{u}_i | i \in S\}$
/* Assign new clustering. */
$P \leftarrow \mathrm{softmax}(\kappa \frac{\mathbf{u}\mathbf{v}^{\top}}{\|\mathbf{u}\|\|\mathbf{v}\|})$
/* Output new features. */
Average pooled feature $\bar{\mathbf{z}} \leftarrow P^{\top}\hat{\mathbf{z}}/P^{\top}\mathbf{1}$
New centroid feature $\mathbf{z} \leftarrow \{\hat{\mathbf{z}}_i | i \in S\}$
Updated centroid feature $\mathbf{z} \leftarrow \mathbf{z} + \mathrm{FC}(\mathrm{LN}(\bar{\mathbf{z}}))$

FPS: Farthest Point Sampling.
MSA: Multi-headed Self-Attention.
FC: Fully Connected Layer.
LN: Layer Norm.  BN: Batch Norm.

**Algorithm 2:** Overall framework

**Input:** Initial segment token $\mathbf{x}_s$, class token $\mathbf{x}_{class}$, position encoding $\mathbf{E}_{pos}$ and grouping steps $\Delta$
**Output:** Feature $\mathbf{f}_{class}$ and $\mathbf{f}_{seg}$
/* Input tokens with priors. */
$\mathbf{z}_0 \leftarrow [\mathbf{x}_{class}; \mathbf{x}_s] + \mathbf{E}_{pos}$
/* ViT with Graph Pooling. */
**for** $l = 1 \ldots L$ **do**
    $\hat{\mathbf{z}}_l \leftarrow \mathrm{ViT\_Encoder}(\mathbf{z}_{l-1})$
    **if** $l \in \Delta$ **then**
       $\mathbf{z}_l \leftarrow \mathrm{GraphPool}(\hat{\mathbf{z}}_l)$;
    **else**
       $\mathbf{z}_l \leftarrow \hat{\mathbf{z}}_l$;
    **end**
**end**
/* Outputs for classification. */
[CLASS] token $\mathbf{f}_{class} \leftarrow \mathrm{LN}(\mathbf{z}_L^0)$
/* Outputs for segmentation. */
Multi-level segment tokens
  $\mathbf{f}_{seg} \leftarrow \mathrm{FC}(\mathrm{BN}([\mathrm{Unpool}(\mathbf{z}_l^{1:n_l}); \ldots])) \quad l \in \Delta$

## 3.2 General and Globalized Backbone Architectures

Most existing vision models digest pixels on a regular grid and update features within a limited range of neighboring grids. Such methods have two limitations: **1)** all features correspond to the same-shape patches in images, and **2)** pixels/patches are locally connected and have little knowledge of global correlation. Yet, segmentations are adaptive to images: every segment has different shapes. Optimal segmentation also requires group-wise correlation among pixels (Shi & Malik, 2000). We thus prefer globalized architectures, such as ViT and GNN (Kipf & Welling, 2016), over the grid-based models.

Specifically, we select ViT as the backbone, which updates features in the context of all inputs without the need for a regular grid. That is, ViT computes features from a globally-connected graph. It consists of multi-headed self-attention modules (MSA) (Vaswani et al., 2017), where features are updated according to pair-wise correlation among all the input features. As a result, ViT encodes global correlation and allows inputs to have different shapes, which is ideal to enable optimal segmentation.

Building upon ViT, our model contains multiple encoder blocks that take segment tokens as inputs: at level $l$, the encoder block updates its inputs $\mathbf{z}_{l-1}$ to $\hat{\mathbf{z}}_l$. Notably, to extract information at different scales, we vary the shapes and sizes of segment tokens throughout the model. Coarser segmentations have fewer segments (tokens), which improves our model efficiency. For the first block, $\mathbf{z}_0$ corresponds to superpixels. For all the other blocks, $\mathbf{z}_{l-1}$ corresponds to segments at different scales.

## 3.3 Hierarchical Groupings of Segment Tokens

Starting with superpixels, we group fine-grained segment tokens into coarse-grained region tokens to obtain more global visual information. With fine-to-coarse segment groupings, we can directly induce hierarchical image segmentations of an input image. As we map superpixels into token features, hierarchical segment groupings become a multi-scale feature clustering and pooling problem. We have two considerations: **1)** The number of coarser segments (tokens) should be freely adjustable during training and inference. **2)** The model should achieve optimal partitioning of inputs. We summarize our feature clustering algorithm, dubbed Graph Pooling Module, in Alg. 1.

Existing methods (Xu et al., 2022; Ke et al., 2022) perform clustering with a set of learnable embeddings, resulting in fixed segmentation granularity. Yet, the optimal number of segments varies with image scales: larger (smaller) images require more (fewer) segments. Instead, we conduct feature clustering with an arbitrary number of centroids that are initialized by sampled inputs. The number of centroids corresponds to the segmentation granularity. We apply the Farthest Point Sampling (FPS) algorithm (Qi et al., 2017) to select a subset of token features as initial centroids. The FPS algorithm

enables the sampled centroids to cover the input feature distributions, unbiased w.r.t. dominant features. We can set the number of clusters flexibly during training and inference.

In particular, we predict the soft clustering assignments $P_l$, which indicates how input features are assigned to sampled centroids. We calculate $P_l$ as softmax-normalized pair-wise similarity among inputs and sampled centroids. We then generate coarsened tokens $\mathbf{z}_l$ by weighted-average aggregating within a cluster. For detailed computation, please refer to Alg. 1.

### 3.4 TRAINING AND INFERENCE

We summarize our overall framework in Alg. 2. We conduct segment grouping at certain levels in the model. We train our CAST using an image-wise contrastive learning framework–Moco-v3 (Chen et al., 2021). The objective is to contrast each image against others in the latent feature space.

To predict classification, we follow MoCo-v3 to apply a 3-layer MultiLayer Perceptron (MLP) head on output `[CLASS]` token. To predict segmentation, we fuse multi-level features as outputs (Hariharan et al., 2015) and unpool higher-level features based on the grouping index w.r.t superpixels.

## 4 EXPERIMENTS

**Datasets**. **1) ImageNet** (Deng et al., 2009) is an object-centric image classification dataset, annotated with $1,000$ object categories–IN-1k. Each image is labeled with one object category, and objects mostly locate at the image center. The training and validation set includes $1.28$M and $50$K images, respectively. We follow Tian et al. (2020) to subsample 100 object categories to create IN-100. The subset consists of 127K and 5K images for training and testing. **2) Pascal VOC 2012** (Everingham et al., 2010) is an object-centric semantic segmentation dataset that contains 20 object categories and a background class. We use the augmented training set (Hariharan et al., 2011) with $10,582$ images and the validation set with $1,449$ images. **3) MSCOCO** (Lin et al., 2014) is a generic scene parsing dataset. The scene contexts are more complex and include more objects in each image (7.3 vs. 2.3) than VOC. Following Van Gansbeke et al. (2021), we train on $118,287$ images of *train2017* split and test on the VOC validation set.

**Model architecture.** For most experiments, we base our architecture on ViT-S (Dosovitskiy et al., 2020), which has $384$ channel dimensions of all encoder blocks. We follow Xiao et al. (2021) to **1)** replace the patch-wise linear projection layer with a stack of four $3 \times 3$ and one $1 \times 1$ convolutional layers; **2)** use $11$, not $12$, encoder blocks to maintain similar model capacity. We adopt the same design choice for our vanilla ViT baselines. For vanilla ViT (our CAST), we set $\mathrm{stride}$ to $2$ among the first four (three) convolutional layers. Our models aggregate pixel features within superpixels, resulting in a similar number of input tokens as ViT baselines. We set graph pooling step $\Delta = \{3, 6, 9\}$ and reduce the number of tokens to $\frac{1}{3}, \frac{1}{6}, \frac{1}{12}$ of initial inputs.

**K-Medoids clustering baselines.** There is no released code for Token Pooling (Marin et al., 2021). We thus re-implement the method by combining PoWER-BERT (Goyal et al., 2020) with K-Medoids clustering algorithm. We follow the same settings as our CAST: we adopt segment tokens and use the same number of tokens at intermediate layers as our CAST.

**Image resolution and token numbers.** For training on all datasets and testing on ImageNet, we set $\mathrm{crop\_size}$ to $224$ and partition 196 superpixels from an image, resulting in around 196 input tokens. For testing on VOC, we generate 1024 (384) superpixels from a $512 \times 512$ input image, resulting in 1024 (384) input tokens for semantic (figure-ground) segmentation. For vanilla ViT, we adopt the same image resolution and use 196 and 1024 input tokens on ImageNet and VOC.

**Model training.** Based on MoCo, we train all models from scratch without any human-labeled supervision. We mostly follow MoCo to set up the hyper-parameters. See Appendix for more details.

**Model testing.** **1) On ImageNet**, we follow MoCo to apply linear probing to evaluate model performance. We freeze the trained model weights and replace the 3-layer MLP head with a randomly initialized linear projection layer as the classifier. We train the classifier with ground-truth labels. **2) On VOC**, we follow Van Gansbeke et al. (2021) to predict semantic segmentation via nearest neighbor search from the labeled VOC training set. We also evaluate performance by fine-tuning models on the training set and testing on the validation set. **3) On figure-ground segmentation**,

following DINO Caron et al. (2021), we binarize multi-head attention maps and select the best binary segmentation among all attention maps for each image. See Appendix for more details.

| backbone | method | # tokens | GFLOPS | IN-100 | IN-1k |
|---|---|---|---|---|---|
| ViT-S | Vanilla | $[196] \times 4$ | 4.67 | 78.1 | **67.9** |
| | K-Medoids | $196, 64, 32, 16$ | **2.84** | 75.8 | 62.8 |
| | Our CAST | $196, 64, 32, 16$ | 3.12 | **78.9** | 66.1 |
| ViT-B | Vanilla | $[196] \times 4$ | 17.8 | 81.7 | - |
| | K-Medoids | $196, 64, 32, 16$ | **10.8** | 81.6 | - |
| | Our CAST | $196, 64, 32, 16$ | 11.7 | **82.0** | - |

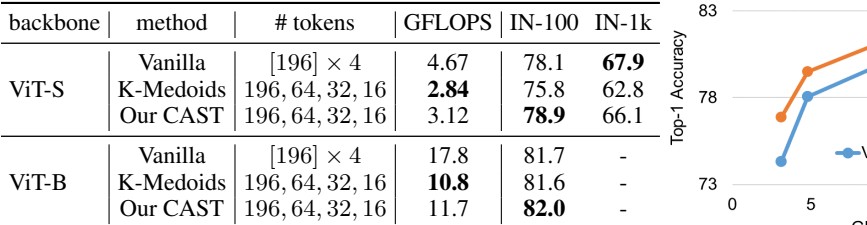

Table 1: Our model achieves a better trade-off between model efficiency and task performance for unsupervised image classification on ImageNet val set. We report the top-1 accuracy of the linear classifier. **Left:** Image classification on IN-100 and IN-1k. Compared to vanilla ViT, our CAST reduces computation overhead by decreasing the number of tokens in the intermediate blocks. **Right:** Image classification on IN-100 val set with different model sizes. Our CAST outperforms vanilla ViT using the same computational budget.

**Result 1: Better accuracy on ImageNet classification.** Our model achieves a better trade-off between model efficiency and task performance for unsupervised image classification. We report top-1 accuracy on both IN-1k and IN-100 (see the left of Table 1). On IN-100, using ViT-S and ViT-B backbone, our CAST achieves comparable performance as vanilla ViT, yet our model is 66.8% and 65.7% computationally more efficient. On IN-1k, our CAST maintains 97.3% performance of ViT baseline (66.1% vs. 67.9% accuracy). Our graph pooling module consistently outperforms K-Medoids clustering. Due to the computation limitations, we set smaller $\text{batch\_size}$ and training $\text{epochs}$ on IN-1K. We also do not apply the pre-training (Marin et al., 2021) or model distillation (Rao et al., 2021) strategy. We can further improve the performance based on such settings.

On IN-100, our model outperforms baselines under different model sizes (see the right plot of Table 1). Our CAST outperforms vanilla ViT using the same computational budget.

| Trained on MSCOCO, to be tuned on VOC | | | before | | after | |
|---|---|---|---|---|---|---|
| method | # tokens | GFLOPS | mIoU | F-score | mIoU | F-score |
| Vanilla ViT | $[1024] \times 4$ | 34.1 | 30.9 | 16.1 | 65.8 | 40.7 |
| K-Medoids | $1024, 320, 160, 80$ | 21.2 | 27.6 | 17.2 | 66.7 | 47.5 |
| Our CAST | $1024, 320, 160, 80$ | 23.6 | **35.9** | **24.7** | **66.8** | **48.1** |

Table 2: Our predicted segmentations are much more precise and better aligned with ground-truth (row 2 column 1 image) semantic boundaries on VOC val set, benchmarked with the regional mIoU and boundary F-score metric. Segmentations are predicted based on segment-wise nearest neighbor retrievals (before fine-tuning, white-colored table columns, row 1 images) and fine-tuned models (after tuning with supervision, gray-colored table columns, row 2 images). Our model achieves better image segmentation with less computation.

**Result 2: Better semantic segmentation on VOC.** We evaluate unsupervised semantic segmentation on VOC in Table 2. We report the regional mean Intersection of Union (mIoU) and boundary F-measure metric. For both segment retrieval and transfer learning, our segmentations are more precise (+5.0% and +1.0% mIoU) and respect object boundaries better (+8.6% and +7.4% F-measure) than vanilla ViT, yet model efficiency is greatly improved. Notably, without applying CRF for post-processing, our model is still able to preserve thin structures (e.g. horse legs) well, whereas ViT results are over-smoothed.

**Result 3: Better figure-ground segmentation on VOC.** We show that our latent attentions capture semantics more precisely. We report the Jaccard similarity between ground truths and predicted foreground masks. Summarized in Table 3, our CAST outperforms DINO and ViT by +2.1%. Our foreground masks can cover dominant foreground objects and better align with object boundaries.

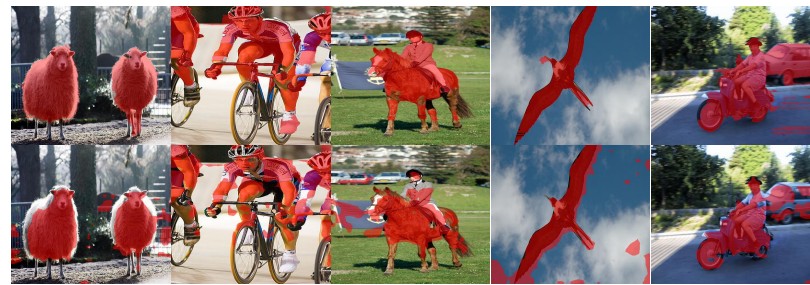

| method | IoU |
| --- | --- |
| Our CAST | **48.0** |
| Vanilla ViT | 45.8 |
| DINO | 45.9 |

Table 3: Our CAST attends to foreground semantics more precisely. All models are trained on IN-1K from scratch. Due to computation limitations, we train vanilla ViT and our CAST with a smaller batch size and fewer epochs. **Left)** Jaccard similarity between ground-truth and predicted foreground masks. **Right)** Visual comparison between our CAST (top row) and ViT (bottom row). Our predicted masks are more coherent within foreground regions and better align with object boundaries.

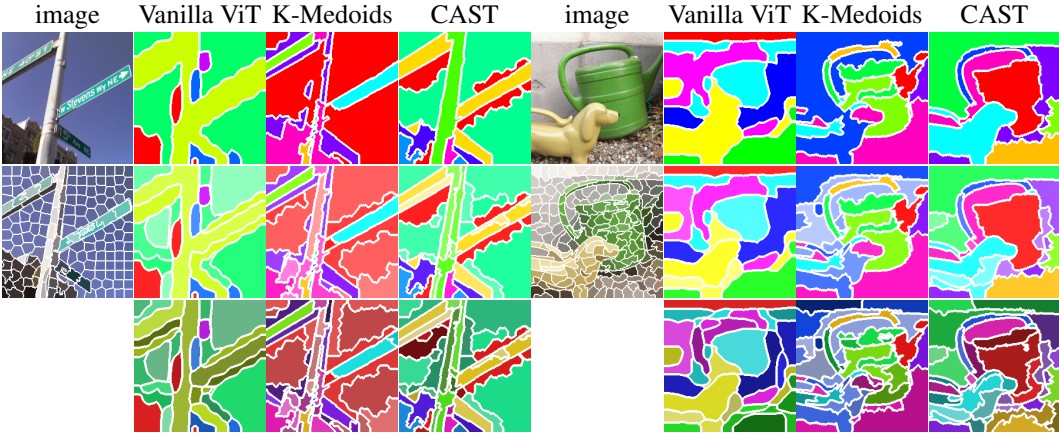

Figure 4: Our CAST generates higher-quality hierarchical segmentation. From left to right, we show the input image and hierarchical segmentations generated by vanilla ViT, K-Medoids clustering, and our CAST. We also show the superpixels (white contours) generated from input images. From top to bottom, we hierarchically segment images into 8, 16, and 32 regions. ViT requires additional pixel-wise clustering (e.g. K-Means) on fixed features, yet the results are over-smoothed. Our CAST naturally generates hierarchical segmentations which capture semantics more precisely.

**Result 4: Better hierarchical segmentation on IN-1K.** Fig. 4 shows that our hierarchical segmentations respect semantics at different levels of granularity better than ViT and K-Medoids baselines. Our model directly produces hierarchical segmentations without needing additional clustering. Bootstrapping from superpixels, our segmentations (e.g. dog regions) align with image contours more precisely than vanilla ViT. Compared to K-Medoids clustering, our model maps similar-semantic fine segments into the same coarse regions (e.g. street-sign pole regions) more consistently..

**Summary.** We develop a novel vision transformer that performs image-wise recognition atop of consistent hierarchical image segmentation, by learning fine-to-coarse features over adaptive segment tokens instead of regular patch tokens. We deliver the first concurrent recognition and hierarchical segmentation model without any supervision, achieving better accuracy and efficiency. The idea can be extended to supervised image classification, with hierarchical semantic segmentation for free.

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

# A    APPENDIX

We develop the first vision transformer that *concurrently* achieves image-wise recognition and hierarchical image segmentation without additional processing. Our CAST outperforms existing token coarsening methods for image classification, semantic segmentation, and attention-induced figure-ground segmentation tasks. In this supplementary, we include more details as followings:

- We present more visual results of hierarchical segmentation on ImageNet in A.1.
- We present more visual results of attention-induced figure-ground segmentations on VOC in A.2.
- We present visual results of attention maps from our CAST on ImageNet in A.3.
- We present the ablation study regarding the proposed graph pooling module in A.4.
- We present the ablation study on inference latency of our CAST in A.5.
- We present the ablation study on effectiveness of superpixels in A.6
- We present quantitative results of hierarchical segmentation on DensePose in A.7
- We illustrate more details of hyper-parameters for training in A.8.
- We illustrate more details of inference in A.9.

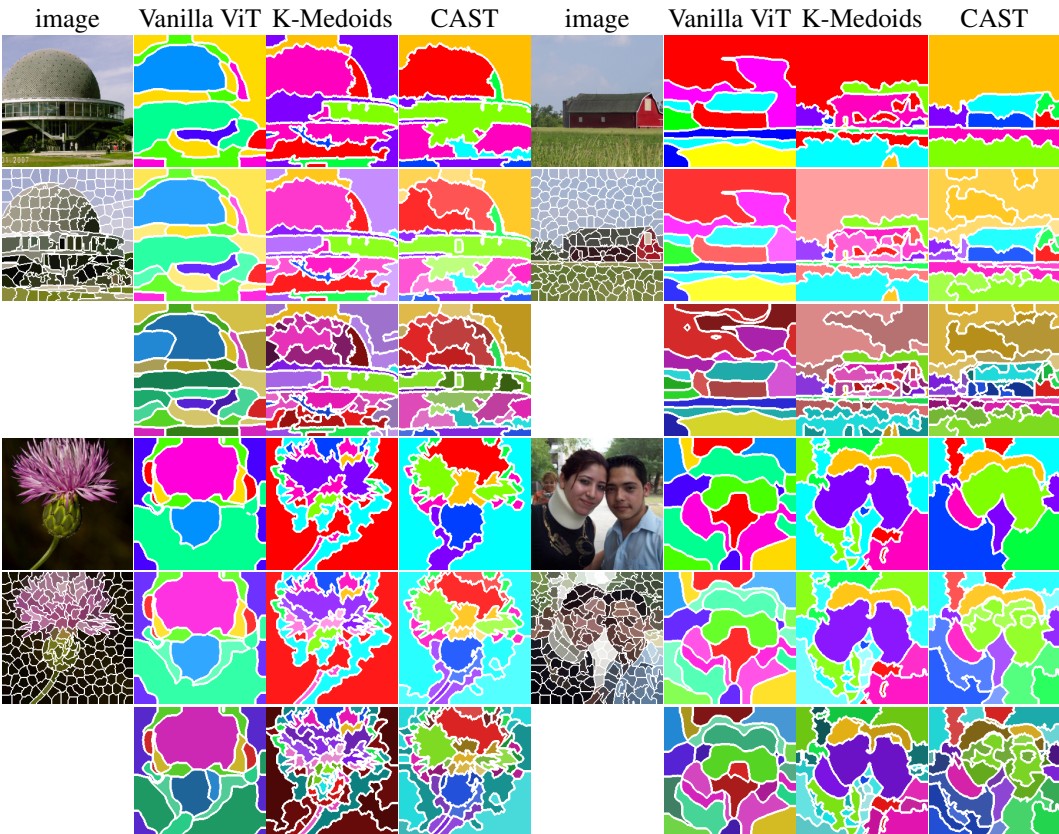

Figure 5: Our CAST generates higher-quality hierarchical image segmentation. From left to right, we show the input image and hierarchical segmentations generated by vanilla ViT, K-Medoids clustering, and our CAST. We also show corresponding superpixels (white contours) generated from input images. From top to bottom, we hierarchically segment the input image into 8, 16, and 32 regions. Vanilla ViT requires additional pixel-wise clustering (e.g. K-Means) on fixed feature representations, yet the results are over-smoothed. Our CAST naturally generates hierarchical segmentations without any post-processing. Our segmentations align with image boundaries and capture semantics more precisely.

CAST (top) vs. Vanilla ViT (bottom)

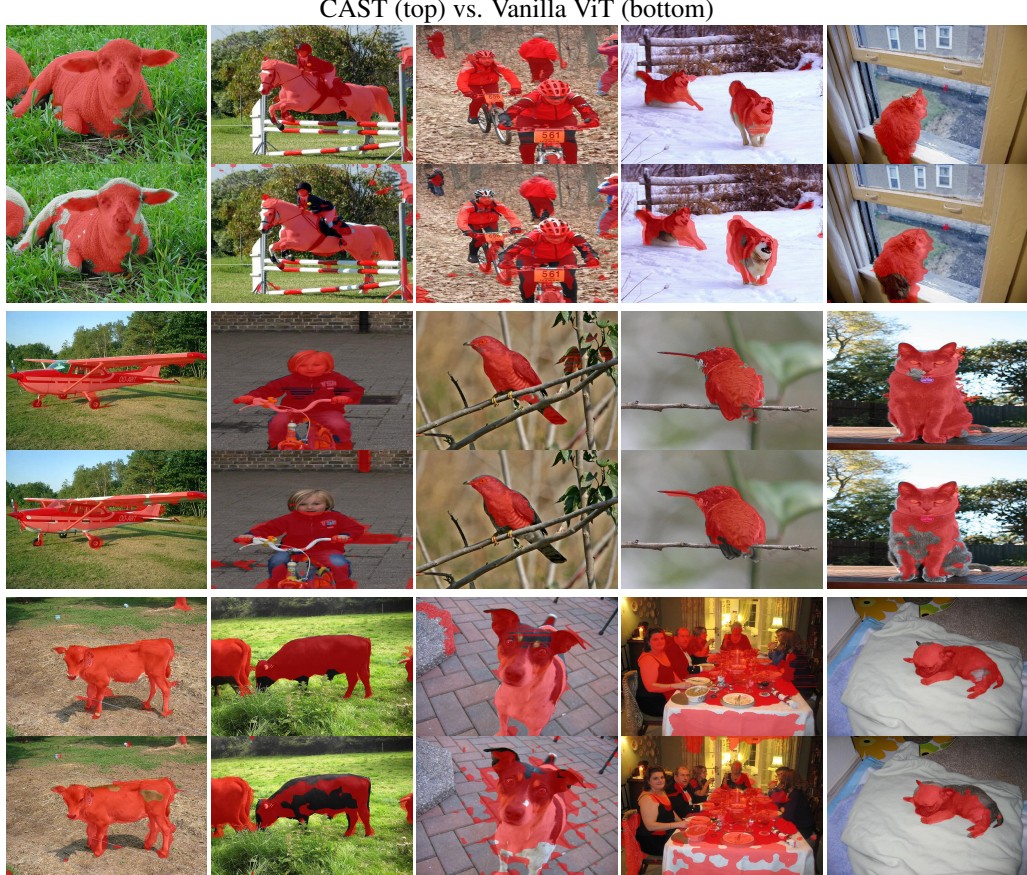

Figure 6: Our CAST (top row) attends to foreground semantics more precisely than vanilla ViT (bottom row) and DINO (Caron et al., 2021) on VOC. We adopt the same procedure as DINO to generate foreground segmentation masks from latent multi-head attention maps. All models are trained on IN-1K dataset from scratch. Our CAST and ViT are trained based on MoCo-v3 (Chen et al., 2021).

### A.1 Visual Results on Hierarchical Segmentation.

We present more visualization results of hierarchical segmentations induced by vanilla ViT, K-Medoids, and our CAST (see Fig. 5). We hierarchically segment the input image into 32, 16, and 8 regions. Notably, Vit baseline requires additional pixel-wise K-Means clustering on fixed feature representations, yet the results are over-smoothed. Our CAST naturally generates hierarchical segmentations without any post-processing. Our multi-scale segmentations align with image boundaries and capture semantics more precisely.

### A.2 Visual Results on Attention-induced Figure-ground Segmentation.

We present more visual results of figure-ground segmentations generated from multi-head attention maps on VOC. Our CAST attends to foreground semantics more precisely than vanilla ViT, and the segmentations preserve object boundaries more accurately. See Fig. 6.

### A.3 Visual Results on Multi-head Attention Maps.

We visualize the multi-head attention maps of the `[CLASS]` token to all the other segment tokens in our vision transformer. As the `[CLASS]` token is optimized for image-wise discrimination, such attention maps indicate the most informative groupings of segments that will induce the most discriminative image-wise representations. We visualize the same attention maps used to generate

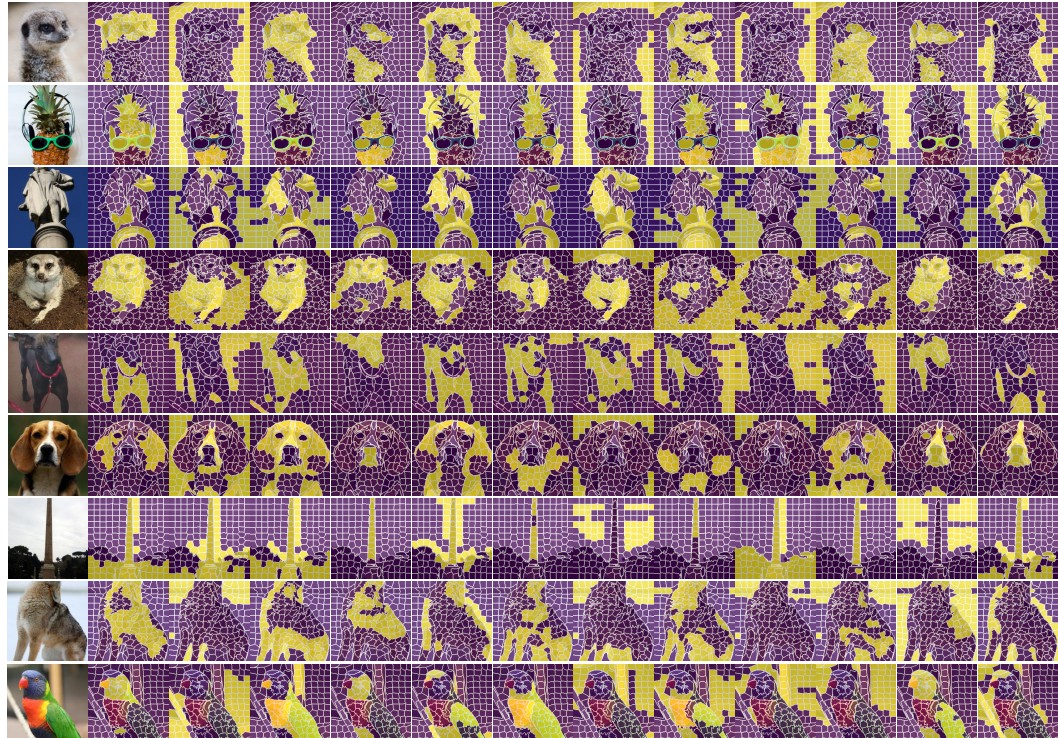

Figure 7: Our multi-head attention maps reveal parts-of-the-whole information of the image on IN-100. **From left to right:** input images and corresponding 12 heads of attention maps of the [CLASS] token to all the other segments. We follow DINO (Caron et al., 2021) to binarize attention maps. We show that the same object parts are together attended in the same head, e.g. face vs. ears vs. nose of the dog. Our model takes segment tokens, resulting in attention maps better aligned with object boundaries.

the figure-ground segmentation, which are the ones in the $9^{th}$ transformer encoder block. The layer takes 32 coarsened segment tokens as inputs, resulting in 12 heads of $32 \times 32$ attention maps. We follow the same procedure as DINO (Caron et al., 2021) to display the binarized attention maps. The threshold is adjusted to keep $60\%$ of the mass. See Caron et al. (2021) for more details.

As shown in Fig. 7, our attention maps reveal parts-of-the-whole information of the image. We observe that the same object parts are together attended in the same attention head, e.g. face vs. ears vs. nose of the dog. It indicates that image-wise recognition requires parts-of-the-whole information. Additionally, our model carries segment, not patch, tokens through the layers, resulting in attention maps better aligned with object boundaries.

| Token | Pooling | Acc. |
|---|---|---|
| Patch | - | 78.1 |
| Segment | - | 78.1 |
| Segment | ✓ | 78.9 |

| Pooling | Acc. |
|---|---|
| Our Graph Pooling | 78.9 |
| K-Medoids: PoWER-BERT | 75.8 |
| K-Means: PoWER-BERT | 73.9 |
| K-Medoids: Random Sampling | 72.3 |

Table 4: Our proposed graph pooling module and loss regularizations improve image classification on IN-100 val set. **Left:** Improved performance by adding each loss regularization. **Right:** Our graph pooling module outperforms K-Medoids and K-Means clustering algorithm by a large margin. Cluster centroids are initiated by either PoWER-BERT or random sampling.

## A.4 Ablation Study on Proposed Graph Pooling Module.

Summarized in Table 4, we demonstrate the efficacy of the proposed graph pooling module. We report top-1 accuracy results on IN-100 dataset. We show that our graph pooling module outperforms K-Medoids and K-Means clustering by a large margin.

| #. of Tokens | Encoder Blocks | GraphPool (FPS) |
|---|---|---|
| 196 | 86.43 | 63.02 (37.64) |
| 64 | 25.4 | 18.2 (9.7) |
| 32 | 12.9 | 9.6 (3.0) |
| 16 | 5 | 6.1 (1.5) |

Table 5: FPS in our graph pool module requires additional computation. We report the inference time (ms) of each module with $384$ channel dimensions and $256$ batch sizes on IN-100. Optimizing the token sampling technique to increase model efficiency is our future work.

## A.5 Ablation Study on Inference Latency

We present the comparison of inference latency among our CAST and vanilla ViT architectures. We report the inference time (ms) of each module with 384 channel dimensions and 256 batch sizes on ImageNet-100. As summarized in Table 5, our graph pool module with FPS indeed requires additional computation. However, our method can also reduce inference time by decreasing the number of tokens in deeper layers.

Disregarding the Conv Stem, vanilla ViT and our CAST take 316.91 and 220.57 ms for inference, respectively. Our model is 30.4% faster than ViT. Optimizing the token sampling technique to increase model efficiency is our future work.

## A.6 Ablation Study on Superpixel

We next verify the superior efficacy of superpixel tokens on dense pixel applications. As shown in Table 6, we compare patch tokens to superpixel tokens on image classification and semantic segmentation tasks. For both ViT baseline and our CAST, we observe significant performance gain for semantic segmentation, yet the performance gap for classification is negligible. We conclude that using superpixels can be very useful in a wide range of dense pixel labeling tasks.

## A.7 Quantitative Results on Hierarchical Segmentation

To demonstrate the efficacy of our hierarchical segmentation results, we compare CAST with K-Medoids on the DensePose dataset (Alp Güler et al., 2018). We process body-part-level labels into the head, torso, upper limb, and lower limb regions. We segment an image into 64, 32, and 16 regions, and evaluate the region-wise coverings (Arbelaez et al., 2010) with ground truths by F1-score. As shown in Fig. 8, our CAST outperforms K-Medoids and is better at capturing semantics at every level of granularity.

## A.8 Hyper-parameters for Training

We list the hyper-parameters for training in Table 7. Mostly follow MoCo, we set $\text{batch\_size}$ to $256$, $\text{learning\_rate}$ to $1.5e^{-4}$, $\text{weight\_decay}$ to $0.1$, and $\text{momentum}$ to $0.9$. We use AdamW (Kingma & Ba, 2014) optimizer. For hyper-parameters of MoCo framework, we set $\text{temperature}$ to $0.2$, output dimension to $256$, $\text{momentum\_coefficients}$ to $0.99$. The 3-layer MLP head has a hidden dimension of 4096. For IN-100, IN-1k and COCO, we set training $\text{epochs}$ to 200, 100 and 400, along with $\text{warmup\_epochs}$ to 20, 10 and 40, respectively. Cosine decay schedule is applied to adjust the learning rate.

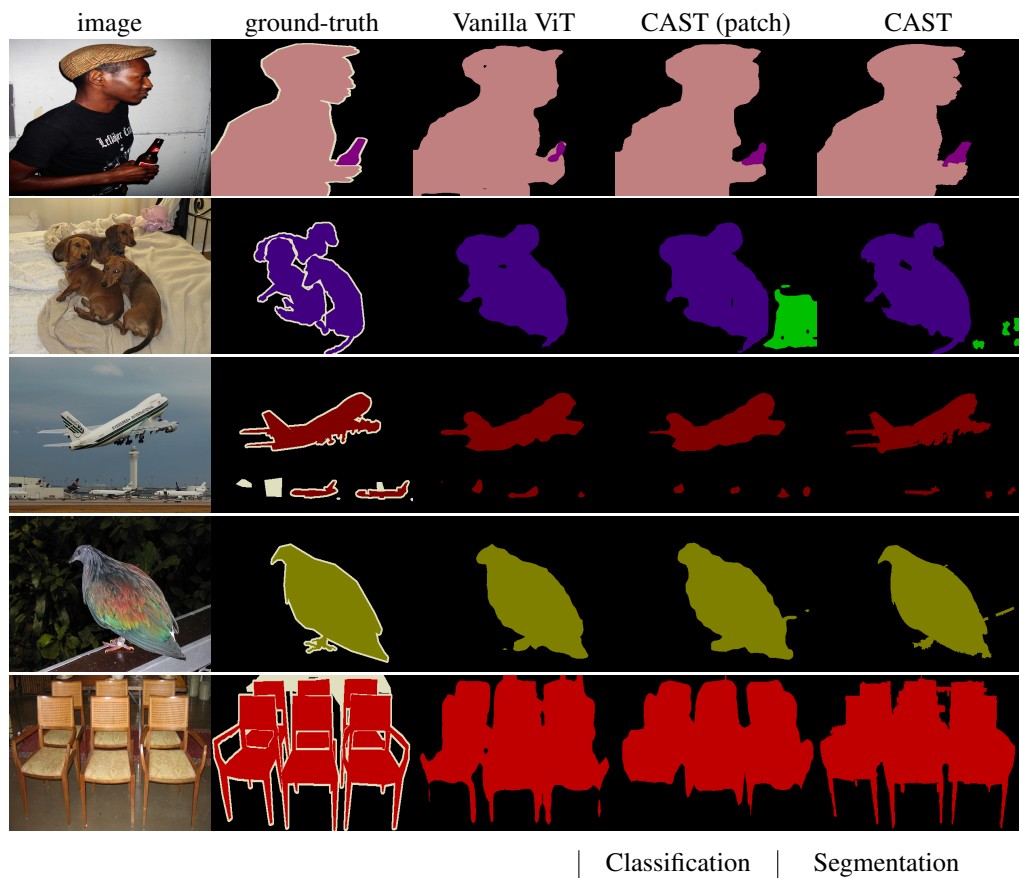

| image | ground-truth | Vanilla ViT | CAST (patch) | CAST |

| Method | GFLOPS | dim | Token | Classification Acc. | Segmentation mIoU | f-score |
|--------|--------|-----|-------|------|------|---------|
| Vanilla ViT | 65.4 | 384 | Patch | 78.1 | 65.8 | 40.7 |
|  |  |  | Segment | 78.1 | 66.5 | 46.7 |
| Our CAST | 42.7 | 384 | Patch | 78.1 | 66.3 | 41.4 |
|  |  |  | Segment | **78.9** | **66.8** | **48.1** |

Table 6: Using superpixel tokens improves the performance of dense pixel applications significantly. **Top:** Visual examples of semantic segmentation by fine-tuned models on VOC val set. **Bottom:** Quantitative results on classification and semantic segmentation tasks. We compare patch tokens to superpixel tokens on image classification (IN-100) and semantic segmentation (VOC). We report the mIoU and boundary f-score performance of semantic segmentation under the setting of transfer learning. We report the GFLOPS of segmentation models, where classification models use the same number of channel dimensions. For vanilla ViT and our CAST, we observe significant performance gain for semantic segmentation, though the performance gap for classification is negligible. In particular, superpixel-based methods preserve thin structures (boundaries) of objects much better than their patch-based counterparts.

## A.9 INFERENCE AND TESTING

For image classification, we apply the linear probing procedure for evaluation. For semantic segmentation, we use the segment retrieval and transfer learning procedure to evaluate model performance.

**Image classification: linear probing.** We follow MoCo-v3 (Chen et al., 2021) to evaluate image-wise discrimination model performance using a linear probing protocol. We freeze the trained model weights and replace the 3-layer MLP head with a randomly initialized linear projection layer as classifier. We train the linear classifier with ground-truth labels and report the top-1 accuracy. Following Chen et al. (2021), we train the linear classifier with 90 epochs on ImageNet dataset. We

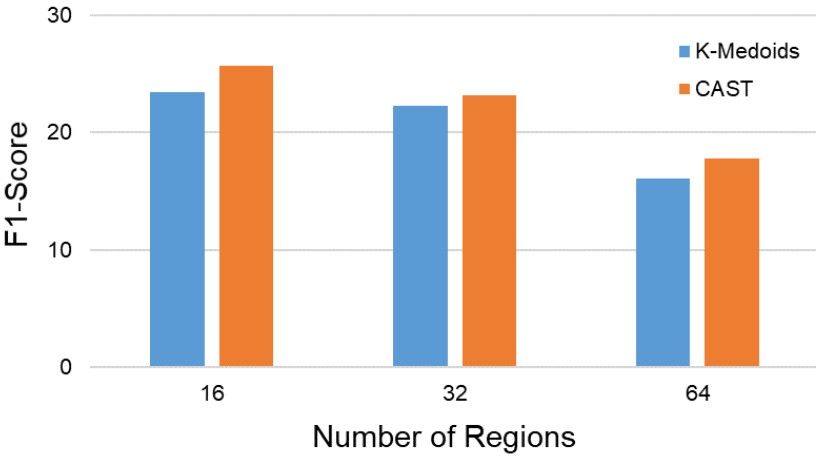

Figure 8: Our CAST delivers better hierarchical segmentation than K-Medoids clustering algorithm. On DensePose (Alp Güler et al., 2018), we parse body-part-level labels into head, torso, upper and lower limb regions. Using CAST or K-Medoids, we partition an image into 64, 32 and 16 regions, and evaluate the region-wise converings with ground-truths by F1-score. Our CAST captures semantics better at every level of granularity.

| Parameter | IN-100 | IN-1K | COCO |
|---|---|---|---|
| batch_size | 256 | 256 | 256 |
| learning_rate | $1.5e^{-4}$ | $1.5e^{-4}$ | $1.5e^{-4}$ |
| weight_decay | 0.1 | 0.1 | 0.1 |
| momentum | 0.9 | 0.9 | 0.9 |
| total_epochs | 200 | 100 | 400 |
| warmup_epochs | 20 | 10 | 40 |
| optimizer | Adam | Adam | Adam |
| learning_rate_policy | Cosine decay | Cosine decay | Cosine decay |
| MOCO : temperature | 0.2 | 0.2 | 0.2 |
| MOCO : output_dimension | 256 | 256 | 256 |
| MOCO : momentum_coefficients | 0.99 | 0.99 | 0.99 |
| MOCO : MLP hidden dimension | 4096 | 4096 | 4096 |

Table 7: Hyper-parameters for training our CAST, K-Medoids clustering, and vanillar ViT on IN-100, IN-1K, and COCO dataset. We follow mostly the same set up as MoCo.

set momentum to 0.9 and weight_decay to 0 for all experiments. On IN-1k, we set batch_size to 1024, learning_rate to 30; on IN-100, we set batch_size to 256, learning_rate to 0.8. SGD is used as the optimizer.

**Semantic segmentation: segment retrieval.** We follow Hwang et al. (2019); Van Gansbeke et al. (2021); Ke et al. (2022) to evaluate semantic segmentation using segment retrieval. We partition an image into several segments and conduct nearest neighbor search to predict the label for each segment. We assign the majority labels from the 20 retrieved segments.

For ViT baselines, we apply the MLP head on each token to generate unit-length output features and upsample the feature maps to the original resolution of the input image. Followed by spherical K-Means clustering algorithm, we partition the image into 36 segments using the output features.

Our CAST does not require additional upsampling and K-Means clustering. For segmentation, our model follows Hypercolumn design (Hariharan et al., 2015) to unpool and fuse multi-level segment tokens. Our model generates the same number of output tokens as the superpixels. We gather pixel features from output tokens based on the superpixel index. Without the need for spherical K-Means

clustering, our CAST predicts an image segmentation using the graph pooling modules. We compute normalized segment features according to such image segmentation.

**Semantic segmentation: transfer learning.** We follow Van Gansbeke et al. (2021) to evaluate model performance using transfer learning protocol. All models are unsupervisedly trained on MSCOCO, and supervisedly fine-tuned on Pascal VOC. We replace the 3-layer MLP head with 2-layer $1 \times 1$ convolutional layers. We set the training steps to 30K, $learning\_rate$ to 0.003, $weight\_decay$ to 0.0001, $batch\_size$ to 16, $crop\_size$ to 512. Following Chen et al. (2016), we adopt poly learning rate policy by multiplying base learning rate by $1 - \frac{iter}{max\_iter}^{0.9}$. We adopt the SGD optimizer. We use only a single-scale image for inference.

