# OpenReview forum: "CAST: Concurrent Recognition and Segmentation with Adaptive Segment Tokens"
_ICLR.cc/2023/Conference — Submitted to ICLR 2023_

### Official Review · Reviewer_5DAX · 2022-10-23

**Confidence:** 4
**Clarity, Quality, Novelty And Reproducibility:** The writing is good and intuitive. No…
**Correctness:** 3
**Technical Novelty And Significance:** 3
**Empirical Novelty And Significance:** 3
**Recommendation:** 6

**Strength And Weaknesses:**

Strength:
1. The motivations to use adaptive segment tokens are natural and straightforward.
2. The paper turns this segmentation problem into a hierarchical segment grouping problem which is very interesting.

Weakness:

1. The paper claims hierarchical segmentation as the main contribution. However, there is no further support for why this is important or useful due to they are outcomes naturally from hierarchical segment grouping.

2. The technical contribution is rather limited. The main technical contribution is the GraphPool algorithm, but further experimental evaluation specifically designed to verify this module's effectiveness is missing.

3. It's mentioned in Sec. 3.3, we have to set the cluster number. Any experiments or investigations on that? Comparison to mode-seeking algorithms such as meanshift may remove the need to set cluster number?

4. Just curious, how can the model learn to correct the over-segmentation error in the superpixel stage? For example, a superpixel contains two or more three semantic different classes. Will the proposed model be able to separate it?



**Summary Of The Paper:**

This paper proposes to use superpixel-based tokens instead of fixed-shape patch tokens for image segmentation, and proposes a graph pooling between transformer blocks to create an increasing size of segmentation from the oversegmented superpixel map.

**Summary Of The Review:**

Overall, this paper poses certain merits in terms of unsupervised image segmentation. However, there are some missing details/problems to answer mentioned in the [Strength And Weaknesses] tab to increase the soundness of the paper.

---

> ### Author Response · Authors · 2022-11-18
> **We thank the reviewer to recognize that our paper turns this segmentation problem into a hierarchical segment grouping problem which is very interesting**
>
> ### Q1. Main contribution? Why hierarchical segmentation is important?
> * Our main contribution is to introduce the first deep learning framework that can tackle both recognition and segmentation concurrently!  Please refer to our detailed response to Question 2 in the general comment.
> * Our framework induces hierarchical segmentation with multiple advantages:
>     1. Though in supervised segmentation, segmentation granularity is determined by predefined labels, overlooking the inherent ambiguity.  Yet, In unsupervised segmentation, what granularity a model should adopt is unclear.  HSG[1] has shown that segmentation consistency between granularities is a powerful cue to discover natural organization.  We adopt the same rationale.
>     2. A practical importance of hierarchical segmentations is that recognition is built upon visual semantics at various levels.  As demonstrated in Table 1 and Table 2, it is capturing this requirement in a self-organizing manner that delivers both higher performance at lower computational costs.
>
> [1] Unsupervised Hierarchical Semantic Segmentation with Multiview Cosegmentation and Clustering Transformers. Ke et al. CVPR 2022.
>
>
> ### Q2. Limited contribution?
> * Our key contribution is to propose the first framework for concurrent recognition and segmentation.  Please refer to our detailed response to Question 2 in the general comment.
> * On the contribution of GraphPool module alone, we have conducted multiple experiments to compare our graph pooling module with K-Medoids clustering [1].  For fair comparison, we adapt K-Medoids clustering to use the similar architecture design: both models are based on ViT-like architectures and take superpixel tokens as inputs.  Our CAST outperforms K-Medoids clustering over classification (Table 1, Table 4), semantic segmentation (Table 2), and hierarchical segmentation (Figure 8).  Please see our paper for more details.
>
> [1] Token Pooling in Vision Transformers. Marin et al. arXiv 2021.
>
> ### Q3. Hand-selected numbers of clusters?
> * The number of tokens is a hyper-parameter in our model.  It is also a common practice for existing deep learning vision models such as CNN, ViT, Swin Transformer, etc. where the number of tokens is fixed for every image.
> * Yes, indeed it would be great to determine the number of tokens automatically given an image, a worthy topic in deep learning in general in the future.
>
> ### Q4. Over-segmentation error?
> * Yes, if a superpixel already contains multiple semantic classes, there is no way to break this superpixel.  However, the impact will be mitigated with other superpixels and contained in some regions, as demonstrated by our benchmark scores.  In short, this could be an issue, but our choice of superpixels is fine enough on these benchmarks.

---

### Official Review · Reviewer_BmNe · 2022-10-24

**Confidence:** 3
**Correctness:** 3
**Technical Novelty And Significance:** 3
**Empirical Novelty And Significance:** 3
**Recommendation:** 6

**Clarity, Quality, Novelty And Reproducibility:**

Overall the paper is good. The motivation and proposed method is clearly presented.

**Strength And Weaknesses:**

Strength

1. The paper is well organized and written so that it is easy to follow and understand.
2. The basic idea is reasonable and insight analysis is convincing.
3. The proposed is technically sound and the novel.

Weaknesses

1. Not sure how the "goodness of grouping and consistency across the hierarchy" are measured and used in proposed method.
2. More experiment with state-of-the-art segmentation and classification methods would be better.

**Summary Of The Paper:**

The paper proposes to concurrently perform image classification and segmentation using adaptive segment tokens, which is demonstrated effective and superior to current methods.

**Summary Of The Review:**

The paper provides in-depth analysis of the human vision for image recognition and segmentation, and a novel method to deal with the task with superior performance.

---

> ### Author Response · Authors · 2022-11-18
> **Thanks for recognizing that our paper is well written and our proposed method is technically sound and the novel**
>
> ### Q1. How to measure and use "goodness of grouping and consistency across the hierarchy"?
> * The goodness of grouping is measured in terms of within-group coherence and between-group distinction, which can be calculated and imposed as a loss on predicted cluster assignment P (see Alg. 1 in our paper).  The consistency across hierarchy is guaranteed through our transformers.
> We have taken Reviewer Jmwj’s suggestion and tried soft cluster assignments instead of our simple hard cluster assignment.  It turns out that soft-assigned graph pooling modules are more effective: the models improve classification accuracy (segmentation mIoU) by 0.5% (3.2%) without using the goodness-of-grouping losses.
> We updated our method by replacing hard assignments with soft assignments in our graph pooling module.  Our method learns to generate optimal clusterings without the need for “goodness of grouping” loss.  Please see Question 1 in the general comment.
>
> ### Q2. Comparison to SOTA methods?
> * Yes, indeed we have conducted additional experiments comparing our method to Swin transformers – the state-of-the-art transformer architectures.  Our method outperforms Swin transformers on unsupervised image classification.  Please see the table in Question 3 in the general comment.

---

### Official Review · Reviewer_Jmwj · 2022-10-25

**Confidence:** 3
**Correctness:** 3
**Technical Novelty And Significance:** 3
**Empirical Novelty And Significance:** 2
**Recommendation:** 5

**Clarity, Quality, Novelty And Reproducibility:**

The paper is fairly easy to understand, although I don't think the writing is of very high quality. The contained research is quite interesting though, I just don't gain any major insights about the key workings of the approach and I'm a bit worried it doesn't really work because it does what one might think at a first glance. Nevertheless, I think the overall idea is fairly novel. Some details are clearly lacking to really make it reproducible and sadly it is unclear if the authors plan to release the code.

**Strength And Weaknesses:**

Strengths:
- The paper proposes an interesting architecture idea and  shows that it has some merit to it with quite some experiments. Especially, the Graph Pooling module seems to work fairly well in their instantiation compared to other baseline approaches.
- Especially that this can be trained without labels makes it rather interesting.

Weaknesses:
- I'm mainly concerned about the training of the proposed model. Specifically the GraphPool block seems to perform a non-differentiable hard one_hot assignment. While I understand that this should be a hard assignment, you cannot propagate gradients to this into the features in such a way that pooling operation learns to pool together meaningful features. As such I also wouldn't really know why the additional losses should be effective and the ablation study also shows that they do not have a major effect on the performance. I would assume this could be fixed with something like a Gumbel softmax. Did you consider doing that? As it is right now, I think the model really only groups together similar superpixels based on similar features that become similar by accident, but not because the features of meaningful supersegments were trained to be more similar. I see this as one of the major weaknesses of this paper. Empirically it performs reasonably well, but it's not really clear if that is because it does what one would expect it to do. I get the same feeling about the superpixel segmentation. As far as I can tell, you apply SEEDS to the RGB image and thus the gradients are passed to the initial feature extraction layers through the regions, but the network can never learn better features to create better superpixel oversegmentations.
- The paper states that the model can deal with varying numbers of input superpixels (and different numbers of coarser supersegments in the later layer) since the number of sampled tokens can just be changed. Sadly there are no real experiments to confirm that. I think this would actually be a super interesting experiment to truly learn something about the architecture. How does it behave when during inference these numbers are actually changed and how important is it that these numbers are selected in the right way? I think focusing a little more on these aspects would make the paper a lot stronger.
- As far as I understand the paper, all of the reported numbers are based on own experiments, including all the baseline numbers. As such I really wonder how well these baselines were tuned. Simply applying the same schedule to some other network doesn't make it an inherently fair comparison. What makes it even worse is that in several cases the margin between the proposed model and the baselines is not even very large, thus raising the question if the model truly performs better than the VIT or k-mediod baselines, or it's just a simple matter of slightly different tuning of the model. To be fair though, the paper does not claim that the method achieves state-of-the-art results. A comparison with external baselines would be very good though!
- The ablations not only show that the additional losses don't make a major difference, but even the fact whether or not superpixels are used instead of patches does not have a major impact on the performance. Did you ever check if the quality of the superpixels matters at all? Could you just use bigger or smaller superpixels?
- Partially the paper is quite informally written and some sections just do not read very well. For example, I have never seen the abbreviation 'a.k.a.' used in a paper and it is consistently missing the last dot. Especially the sections dealing with training setups are not very nice to read and I think they would be better placed in the appendix, likely in tabular form. In general another pass over the paper, checking some spelling and grammar, would likely be good.

Question:
- A quick google search yielded the following CVPR'22 paper "Semantic Segmentation by Early Region Proxy". It shows that there is at least one other transformer-based method using superpixels. Even though I don't think it's extremely related, it could make a good addition to the related work.


**Summary Of The Paper:**

The paper proposes to replace the standard rectangular patch based tokens in transformers with superpixel based regions. Features for each superpixel are pooled to a token and these are used in a hierarchical transformer architecture. When reducing the number of tokens (GraphPool), furthest point sampling is used to select a subset of tokens and all the tokens from the previous layers are grouped to the sampled tokens with the highest similarity. The complete network is trained in an unsupervised fashion, meaning that a hierarchical segmentation can be extracted from the network without any labels. Using some additional supervision on top of this results in an approach that can perform semantic segmentation in a hierarchical way. The overall idea to modify a transformer architecture to obtain a hierarchical segmentation without the need for direct segmentation supervision is very interesting.

**Summary Of The Review:**

Overall I like the core concept automatically learning a hierarchical segmentation without the need for labels, however, I have some doubts about how the model really works and how well it works compared to external baselines. I think adding some more experiments regarding these issues and giving the paper another good polishing will likely make it a lot better. As such I'm a little unsure whether to accept this or not. Depending on the other reviews and the rebuttal, I'm willing to change my opinion more towards the positive side though.

---

> ### Author Response · Authors · 2022-11-18
> **We than the reviewer for acknowledging that our proposed architecture is interesting and has some merit with quite some experiments**
>
> ### Q1. Concerns on the hard cluster assignments
> * Great suggestion! In fact, our key contribution is a single framework that performs concurrent recognition and segmentation, not the technicality of hard cluster assignments.  Our graph pooling module is compatible with either hard or soft assignments, while hard assignments are just conceptually simpler and easier to implement.   It turns out that soft-assigned graph pooling modules are more effective: the models improve classification accuracy (segmentation mIoU) by 0.5% (3.2%) without using the goodness-of-grouping losses.  Please see the tables in Question 1 in the general comment.
> * Yes, indeed learning to generate superpixels that better reflect desired groupings could potentially improve the performance further.  Currently, we adopt the fixed SEEDS method for both conceptual and implementation simplicity, and we leave an end-to-end learnable framework for future exploration (by us or other researchers in the community).
>
> ### Q2. Varying numbers of input superpixels and intermediate segments?
> * Yes!  We have conducted the requested experiments (results below) and our findings are very intuitive:  Use more superpixels for finer details and smaller objects.  We will include such results in the final version.  Thanks for the suggestion!
> On VOC, we use the same 1024 input tokens and vary the number of output tokens.  We report mIoU performance on images with large and small objects, respectively.
>
> **Segmentation performance with varying numbers of output tokens**
> | # output tokens | Large objects | Small objects |
> | :-----------: | :-----------: | :-----------: |
> | 20 | 44.22 | 32.69 |
> | 40 | 43.62 | 33.66 |
>
> * The number of tokens adopted by baselines is 196 for ImageNet and 1024 for VOC.  To investigate the sensitivity of our method to the number of tokens, we increase the number of tokens from our 196 to 384 for ImageNet and decrease our 1024 to 196 for VOC.  The results show that there is a slight drop of performance.  Since the change of # tokens only happens during inference, our explanation for the poorer classification result is that the model has not seen tokens at a finer resolution during training for classification; whereas it is very intuitive that the segmentation performance gets worse due to loss of spatial resolution in individual tokens.  Given the drastic change in the number of tokens, 2x more tokens for classification and 5x fewer tokens for segmentation, the only slight drop in performance is actually rather remarkable, reflecting the robustness of our approach without any training or adaptation.
>
> **Performance with varying numbers of input tokens**
> | Dataset | # input tokens  | Performance |
> | :-----------: | :-----------: | :-----------: |
> | ImageNet-100 | 196 -> 384 | 78.92 -> 77.78 |
> | VOC | 1024 -> 196 | 35.9 -> 34.3 |
>
> ### Q3. Sub-optimally tuned baselines?
> * We strictly follow MoCo-v3 to set up our hyper-parameters for training.  Due to our hardware limitation (a smaller compute infrastructure with eight 12GB Nvidia TitanX cards), we can only decrease the batch_size from 1024 to 256 and vary the training epoch w.r.t each dataset.  Other parameters are the same as MoCo-v3.
> * Our method outperforms Swin transformers on unsupervised image classification.  Please see the table in Question 3 in the general comment.
>
> ### Q4. Quality of superpixels?
> * Yes, the quality of superpixels matters.  Using superpixels generated by SEEDS improves classification and segmentation performance by 1.5% accuracy and 2% mIoU compared to SLIC.  We leave an end-to-end learnable framework for future exploration.
> * The granularity of superpixels is flexible and can be adjusted.  Please refer to detailed discussions (to Question 2) about our additional experiments on varying the number of input tokens during inference only.  However, in our paper, we choose the same number of superpixels used in our CAST as the number of patches used in ViT baselines for fair comparison.
>
> ### Q5. Informal abbreviations?
> * Great suggestion!  We’ve updated our draft to get rid of such informal abbreviations, adopted the suggested tabular form, and corrected typos.  There is always room for improvement and we will continue to polish the writing till final publication.
>
> ### Q6. Related works on superpixels?
> * Yes, we’ve updated our draft to create an additional paragraph in Related Work about superpixels and also included this suggested reference.

---

### Official Review · Reviewer_kF9A · 2022-10-29

**Confidence:** 3
**Correctness:** 3
**Technical Novelty And Significance:** 3
**Empirical Novelty And Significance:** 3
**Recommendation:** 6

**Clarity, Quality, Novelty And Reproducibility:**

The paper is quite clear, and well organized. The method is a novel combination of traditional segmentation with transformer methods, addressing a limitation of transformers in using fixed tokens. There appears to be sufficient detail to reproduce.

**Strength And Weaknesses:**

Strengths

Generalizing vision transformers to incorporate tokens of arbitrary shape and size is a significant contribution. As the authors state, partitioning an image into fixed size and shape tokens is an artificial construct that ignores image content. Removing that constraint is a worthy goal and should lead to improved performance.

Constructing the model to handle arbitrary image regions requires generalizing it to handle a variable number of tokens. Fig. 2 is a very clear, effective illustration of the important aspects of transformer-based segmentation algorithms and the advantages of the proposed approach. Explicitly computing and leveraging image regions presents complexities but is much more faithful to the image.

The method is based on hierarchical segmentation using superpixels, an interesting revisitation of traditional methods combined with transformers. Before deep learning such methods were popular and achieved SOTA performance on segmentation problems, and combining these techniques with deep learning is a worthwhile attempt. The primary benefit, as shown in the results, is an improvement in computational efficiency rather than overall accuracy. The results show modestly improved accuracy over the VIT baseline in most cases, with significant improvements in computational cost.

Combining region segmentation and image recognition through jointly optimizing them is a compelling idea, and has been treated only lightly in recent years.

Weaknesses

On page 1 the authors state that “Models optimized for image classification have no concepts of parts and wholes; recognition is achieved without understanding how different parts such as eyes and face are organized for the whole animal.” This claim seems hard to justify, as CNNs are designed to learn object parts and their relationships, and many works have shown this to be true. This statement should be qualified appropriately or softened considerably.

The results do not show a significant gain in segmentation accuracy from the proposed method compared to baseline VIT using standard (square) image tokens, on ImageNet, which is disappointing. However the primary strength of the method is in fine-grained accuracy, and this performs better as shown in tables 2 and 3 for VOC.


**Summary Of The Paper:**

The paper proposes a method for jointly computing region segmentation and image recognition, with any supervision to learn segmentations. The concept is based on hierarchical region segmentation to feed tokens into an image transformer, instead of fixed-size image patches. Experiments show that the method achieves recognition accuracy comparable to the baseline transformer but at half the computational cost, and considerable increases in segmentation accuracy.

**Summary Of The Review:**

The paper is well written, with a novel idea that is validated through thorough experiments. It should be of interest to those working in image segmentation and recognition.

---

> ### Author Response · Authors · 2022-11-18
> **Thanks for recognizing that our idea to generalize vision transformers to incorporate tokens of arbitrary shape and size is a significant contribution**
>
> ### Q1. Previous methods often ignore parts-and-wholes relationships?
> * Yes, we will elaborate on how current CNNs fall short of characterizing parts-and-whole relationships.  For example, CNNs tend to latch onto discriminative parts [1,2] such as faces, often missing inconspicuous body parts that go with the face.  Our work models parts-and-wholes relationships more explicitly through learning transformers that map fine-grained segments to coarse-grained segments.
>
> [1] Learning deep features for discriminative localization. Zhou et al. CVPR 2016.
>
> [2] Grad-cam: Visual explanations from deep networks via gradient-based localization. Selvaraju et al. ICCV 2017.
>
> ### Q2. Comparable segmentation results on VOC and sub-optimal classification accuracy on IN-1K?
> * Yes, indeed our previous CAST results are far better (+8%) than ViT  in boundaries although comparable (+2%) per the crude metric mIoU.  However, our new results obtained by merely replacing hard cluster assignments with soft assignments achieve far better (+5%) mIoU and better efficiency (+31%) on unsupervised VOC segmentation.  On ImageNet-1K, our CAST maintains 97% performance but uses only 66% computation compared to vanilla ViT.  We can further improve our performance by using the similar model size as vanilla ViT.
>
> | VOC (before fine-tuning) | # tokens |  GFLOPS | mIoU |
> | :-----------: | :-----------: | :-----------: | :-----------: |
> | ViT-S | 1024 x 11 | 34.1 | 30.9 |
> | Ours | 1024 x 3, 320 x 3, 160 x 3, 80 x 2 | 23.6 | 35.9 |

---

### Author Response · Authors · 2022-11-18
**We thank reviewers for recognizing that our idea to generalize transformers to incorporate segment tokens and to formulate a hierarchical segment grouping problem is very interesting and a significan contribution**

We thank reviewers for pointing out that “The paper turns this segmentation problem into a hierarchical segment grouping problem which is very interesting”, “Generalizing vision transformers to incorporate tokens of arbitrary shape and size is a significant contribution”, “The basic idea is reasonable and insight analysis is convincing”, and “the Graph Pooling module seems to work fairly well in their instantiation compared to other baseline approaches”.  We are cleaning up our codes and plan to release the code base when it is ready.  Now we address first common and then individual concerns.

### Q1. Efficacy of proposed losses: goodness of grouping?
* We investigate whether the hard cluster assignment in our Graph Pooling can be relaxed to soft assignment.  It turns out that soft-assigned models do not need additional losses and outperform hard-assigned ones on both classification and segmentation tasks.  We can thus replace hard assignments with soft assignments and get rid of the "goodness of grouping" loss altogether.


**On ImageNet-100:**

| Assignments     |  Goodness of grouping | Accuracy |
| :------------ | :-----------: | :-----------: |
| Hard     | x       | 78.0 |
| Hard     | v       | 78.4 |
| Soft     | x       | 78.9 |
| Soft     | v       | 78.4 |

**On VOC (before fine-tuning):**
| Assignments   | mIoU |
| :------------ | :-----------: |
| Hard     | 32.7 |
| Soft     | 35.9 |


### Q2. Main contribution?
* Our main contribution is to introduce the first deep learning framework that can tackle both recognition and segmentation concurrently!  By adopting superpixels, we generalize transformers to incorporate tokens of arbitrary shape and size.  By adopting graph pooling modules, our model hierarchically generates feature representations along with corresponding image segmentations.  Our CAST does not need after-fact pixel-wise clustering!
Moreover, our design choices bring multiple advantages.  Adaptive segment tokens with graph pooling modules 1) directly produce hierarchical image segmentation, 2) increase the model efficiency, and 3) provide free and better object-centric segmentations.
Our method is orthogonal to existing SOTA hierarchical vision transformer architectures, which presume images as grid-structured data.  Our paper takes a more general perspective of images as graphs.  Our idea can be easily generalized to other non-grid data and visual processing of images at later stages.


### Q3. Comparison to other SOTA methods?
* We demonstrate the efficacy of our method by additionally comparing it with Swin transformer (state-of-the-art transformer architecture) on unsupervised image classification.  We select Swin-Tiny which has similar computational efficiency as ViT-S.  We report top-1 accuracy on ImageNet-100.  All models are trained with the same hyper-parameters mentioned in the paper.  **Without using any labels, our CAST outperforms both ViT-S and Swin-Tiny with less computation.**

| IN-100     |  ViT-S | Swin-Tiny | Ours |
| :------------ | :-----------: | :-----------: | :-----------: |
| Acc.     | 78.1       | 78.3 | 78.9 |
| GFLOPS     | 4.67       | 4.50 | 3.12|

---

### Decision · Program_Chairs · 2023-01-20

**Decision:**

Reject

**Justification For Why Not Higher Score:**

- Justification of the choices in the proposed framework
- Lack of end-to-end-training from super-pixels
- Consolidated experiments

**Justification For Why Not Lower Score:**

N/A

**Metareview: Summary, Strengths And Weaknesses:**


This paper introduces a method for Concurrent recognition and segmentation with Adaptive Segment Tokens (CAST). From an off-the-shelf unsupervised superpixel generator (SEEDS), each superpixel is considered as a token which feature is computed by average pooling of a convolutional backbone. The authors introduce a graph pooling approach to obtain a hierarchical image representation. The model is trained in a unsupervised manner from a self-training method (MOCO), and can jointly perform image recognition and multi-scale image segmentation. Experiments are conducted on ImageNet, MSCOCO and PASCAL-VOC, overall showing comparable performances with respect to the ViT baseline at smaller computation cost, and better segmentation performances. \
The paper initially received three borderline accept recommendations and one borderline reject recommendation. The main concerns pointed out by reviewers related to the lack of novelty of the approach, the need to clarify the end-to-end training for non-differentiable blocks, the significance of the experiments and the comparison to stronger baselines. During the rebuttal, the authors provided a variant of the approach with a soft assignment, which enables to boost performances and to get rid of the "goodness of grouping" loss. They also provided on new comparison with the Swin baseline. \
The AC carefully reads the submission and the discussions. Since it was a borderline paper, the AC organized a virtual meeting with the reviewers (see details below). During the virtual meeting RJmwj was not fully satisfied by the rebuttal and recommended weak rejection. R5DAX also raised concerns about the answers and the limitation of the proposed approach, but was still slightly positive about the submission. RBmNe was not present in the virtual meeting but participated to the discussion: he was not convinced by the rebuttal and chose to decrease his grade.

The AC considers that the idea of defining transformers over token of irregular shape is interesting, and that the proposed multi-resolution graph pooling approach is sound. However, a main limitation remains the fixed input super-pixels, and the approach would be much stronger with an end-to-end learning strategy. The experiments could also be pushed and deepened to show the practical interest of the proposed approach. Also, as discussed on the  AC-reviewer meeting, the modification on the paper (soft assignment and goodness of grouping loss) has not been fully described on the revised paper, and this modification would need another round of review. Therefore, the AC recommends rejection but highly encourages the authors to re-submit their works based on reviewers' feedback.


**Summary Of Ac-Reviewer Meeting:**

RJmwj considered that the rebuttal was very detailed but was not fully satisfying. He considered that the answers often follow the easy way, by providing straight answers or new experiments but often avoiding the core of the issues' content. For example, to fulfil RJmwj's request, the authors perform a variant of their method using soft-assignment, which proves to be more effective and enables to relax the need of the goodness of grouping loss. RJmwj considered that there were several options to perform this soft-assignement that need to be discussed and justified. \
R5DAX also shared the same feeling, e.g. on the answer related to the number of clusters, where the answer was limited to the statement that it is a common practice to setup it to a fixed value, without discussing its potential impact on the approach. The same feedback was given for the answer about the impact of over-segmentation error. This said, R5DAX was  overall still slightly positive about the paper.\
RBmNe was not present in the virtual meeting but participated to the discussion: he was not convinced by the rebuttal, and considered that his concern on novelty has not been properly addressed, and that the proposed approach limits to a combination of existing methods.\
Without negating the the paper's merits, there was a consensus among active reviewers that the submission would highly benefit from a full re-writing and a new round of review.